# ISG15 counteracts *Listeria monocytogenes* infection

Lilliana Radoshevich[1,2,3]*, Francis Impens[1,2,3], David Ribet[1,2,3], Juan J Quereda[1,2,3], To Nam Tham[1,2,3], Marie-Anne Nahori[1,2,3], Hélène Bierne[4], Olivier Dussurget[1,2,3,5], Javier Pizarro-Cerdá[1,2,3], Klaus-Peter Knobeloch[6], Pascale Cossart[1,2,3]*

[1]Unité des Interactions Bactéries Cellules, Institut Pasteur, Paris, France; [2]Institut National de la Santé et de la Recherche Médicale, U604, Paris, France; [3]Institut National de la Recherche Agronomique, USC2020, Paris, France; [4]Institut National de la Recherche Agronomique, UMR1319, Micalis, AgroParisTech, Jouy-en-Josas, France; [5]Université Paris Diderot, Sorbonne Paris Cité, Cellule Pasteur, Paris, France; [6]Molecular Genetics Group, Neuropathologie, Universitätsklinikum Freiburg, Freiburg, Germany

**Abstract** ISG15 is an interferon-stimulated, linear di-ubiquitin-like protein, with anti-viral activity. The role of ISG15 during bacterial infection remains elusive. We show that ISG15 expression in nonphagocytic cells is dramatically induced upon *Listeria* infection. Surprisingly this induction can be type I interferon independent and depends on the cytosolic surveillance pathway, which senses bacterial DNA and signals through STING, TBK1, IRF3 and IRF7. Most importantly, we observed that ISG15 expression restricts *Listeria* infection in vitro and in vivo. We made use of stable isotope labeling in tissue culture (SILAC) to identify ISGylated proteins that could be responsible for the protective effect. Strikingly, infection or overexpression of ISG15 leads to ISGylation of ER and Golgi proteins, which correlates with increased secretion of cytokines known to counteract infection. Together, our data reveal a previously uncharacterized ISG15-dependent restriction of *Listeria* infection, reinforcing the view that ISG15 is a key component of the innate immune response.

*For correspondence: lilliana. radoshevich@pasteur.fr (LR); pascale.cossart@pasteur.fr (PC)

## Introduction

*Listeria monocytogenes* is a food-borne pathogen that can cause enteric infections. In addition, in immunocompromised individuals it can cross the blood–brain barrier and in pregnant women the feto-placental barrier potentially leading to cases of meningitis and septicemia. To be fully virulent, *Listeria* must evade macrophage killing, enter and replicate in epithelial cells and spread from cell to cell. Towards these aims *Listeria* subverts a number of normal host cell functions in order to promote its own replication and dissemination through a plethora of well-characterized virulence factors (*Cossart and Lebreton, 2014*). Conversely, *Listeria* induces a rapid and sterilizing CD8[+] T cell-mediated adaptive immune response that has been extensively characterized (*Lara-Tejero and Pamer, 2004*; *Pamer, 2004*). A more recent area of investigation has been the innate immune response to the pathogen (*Stavru et al., 2011*).

Since *Listeria* is able to survive and replicate in the cytosol, several groups have sought to elucidate how bacteria are sensed within macrophages and more recently within nonphagocytic cells. Once *Listeria* has escaped from the phagosome, its multidrug efflux pumps secrete small molecules leading to activation of an IRF3-dependent cytosolic surveillance pathway (CSP), resulting in type I interferon activation (*Crimmins et al., 2008*). One of these small molecules, cyclic-di-AMP, is sufficient to activate interferon β production in macrophages (*Woodward et al., 2010*). In nonphagocytic cells,

**eLife digest** *Listeria monocytogenes* is a bacterium that can cause serious food poisoning in humans. Infections with this bacterium can be particularly dangerous to young children, pregnant women, the elderly, and individuals with weakened immune systems because they are more susceptible to developing serious complications that can sometimes lead to death.

The bacteria infect cells in the lining of the human gut. Cells that detect the bacteria respond by producing proteins called interferons and other signaling proteins that activate the body's immune system to fight the infection. One of the genes that the interferons activate encodes a protein called ISG15, which helps to defend the body against viruses. However, it is not clear what role ISG15 plays in fighting bacterial infections.

Here, Radoshevich et al. studied the role of ISG15 in human cells exposed to *L. monocytogenes*. The experiments show that ISG15 levels increase in the cells, but that the initial increase does not depend on Interferon proteins. Instead, ISG15 production is triggered by an alternative pathway called the cytosolic surveillance pathway, which is activated by the presence of bacterial DNA inside the cell.

Further experiments found that ISG15 can counteract the infections of *L. monocytogenes* both in cells grown in cultures and in living mice. ISG15 modifies other proteins in the cell to promote the release of proteins called cytokines that help the body to eliminate the bacteria. Radoshevich et al.'s findings reveal a new role for ISG15 in fighting bacterial infections. A future challenge will be to understand the molecular details of how ISG15 triggers the release of cytokines.

type I interferon induction seems to emanate from sensing of triphosphorylated RNA molecules via a RIG-I and MAVS-dependent pathway (*Abdullah et al., 2012*; *Hagmann et al., 2013*). Type I interferon production subsequently leads to autocrine or paracrine activation of interferon-stimulated genes (ISGs). We have recently shown that *Listeria* also activates the type III interferon pathway (*Lebreton et al., 2011*; *Bierne et al., 2012*), a pathway which was discovered much later than type I interferon (*Kotenko et al., 2003*; *Sheppard et al., 2003*). The type III interferon receptor has a more limited tissue expression pattern than the receptor for type I interferon but activates a signaling pathway similar to that of the type I interferon receptor. Several laboratories including ours have recently contributed to the understanding of the type III interferon-dependent response to intracellular viral and bacterial infections. Strikingly, the type III response occurs via peroxisomal MAVS (*Dixit et al., 2010*; *Odendall et al., 2014*).

The role of one particular ISG, ISG15, during bacterial infection remains elusive. ISG15 is a linear di-ubiquitin-like molecule (ubl) that is conserved from zebrafish to human; however, it is much less well characterized than other ubls (*Bogunovic et al., 2013*). It can conjugate to over 300 cellular proteins and can also function as a cytokine to induce interferon-$\gamma$ production in peripheral blood mononuclear cells (*D'Cunha et al., 1996*; *Giannakopoulos et al., 2005*; *Zhao et al., 2005*). Since *Listeria*, as other pathogenic bacteria, often targets post-translational modifications during infection (*Bonazzi et al., 2008*; *Ribet and Cossart, 2010*; *Ribet et al., 2010*), we were interested in investigating the interplay between the interferon-stimulated ubl ISG15 and *Listeria*. ISG15 plays an important role in the innate immune response to viruses. *Isg15* expression becomes rapidly upregulated, and the protein is subsequently conjugated to cellular and/or viral targets following type I interferon induction (*Zhang and Zhang, 2011*). Mice deficient in ISG15 are susceptible to infection with Influenza, Sindbis, and Herpes viruses (*D'Cunha et al., 1996*; *Lenschow et al., 2005*, *2007*). Furthermore, many viruses encode proteins that specifically impair ISGylation (*Frias-Staheli et al., 2007*). ISG15 seems to be unique among ubls, as it can both modify specific target proteins and non-specifically modify proteins cotranslationally (*Frias-Staheli et al., 2007*; *Durfee et al., 2010*; *Zhao et al., 2010*). Since ISG15 is strongly induced by type I interferon, which is produced during bacterial infection, we aimed to decipher whether ISG15 is induced during *Listeria* infection and if so whether ISGylation acts as a means of host defense against invading bacteria.

Here, we show that in nonphagocytic cells ISG15 is dramatically induced upon *Listeria* infection and that, surprisingly, early induction can be type I interferon independent. *Listeria*-mediated ISG15 induction depends on the CSP, which senses bacterial DNA and signals through STING, TBK1, IRF3, and IRF7. Most importantly, we demonstrate that ISG15 counteracts *Listeria* infection both in vitro and

in vivo. We identified protein targets of ISGylation following overexpression of ISG15 using stable isotope labeling in tissue culture (SILAC) analysis and uncovered a prominent enrichment in integral membrane proteins of the endoplasmic reticulum and Golgi apparatus. This enrichment correlated with an increase in canonical secretion of cytokines known to control infection, highlighting a new mechanism of regulation of the host response to an intracytosolic pathogen.

## Results

### ISG15 is induced by *L. monocytogenes* infection both in vitro and in vivo

To test whether ISG15 and ISGylation are induced upon bacterial infection, we infected HeLa cells with *Listeria*. Upon *L. monocytogenes* infection, ISG15 was massively induced, whereas incubation with the related non-pathogenic bacterium, *Listeria innocua*, did not lead to an increase in ISG15 production (*Figure 1A*). We subsequently monitored ISG15 expression in cells infected with *Listeria* over time. ISG15 protein levels increased relatively rapidly; the unconjugated protein was already present at 6 hr post infection and accumulated steadily throughout the infection (*Figure 1B*). We next investigated whether ISG15 induction following *Listeria* infection also occurs in vivo. After 72 hr of systemic sub-lethal *Listeria* infection in mice, there was a robust induction of ISG15 and ISGylated conjugates in infected liver tissue, revealing that *Listeria* infection leads to ISG15 induction both in vitro and in vivo (*Figure 1C*).

### ISG15 induction can be type I interferon independent

Since ISG15 protein levels increased relatively rapidly and *ISG15* is known to be transcriptionally induced in response to interferon, we monitored transcript levels of both *ISG15* and *IFNB1* by quantitative real time PCR (qRT-PCR). Interestingly, we found that the two transcripts are concomitantly induced after 3 hr of infection with *Listeria* (*Figure 1D,E*). This concomitant induction led us to hypothesize that *ISG15* could be induced in an interferon-independent manner during *Listeria* infection. In order to test this hypothesis, we used the viral protein B18R to block signaling from the interferon receptor (*Chairatvit et al., 2012*). The protein acts similarly to a blocking antibody. When cells are pretreated with B18R, the viral protein inhibits binding of interferon to its receptor, which is thus prevented from signaling. Following pretreatment with B18R, HeLa cells were either stimulated with interferon or infected with *Listeria* to assess whether the bacterial-ISG15 induction was dependent on secreted interferon signaling in an autocrine or paracrine manner. We observed that bacteria-induced ISG15 production was not diminished by B18R pretreatment in stark contrast to the interferon-induced ISG15 signal, which was almost entirely abrogated by B18R pretreatment (*Figure 1F*). To confirm the B18R results, we took advantage of a human fibrosarcoma cell line, 2fTGH, from which interferon-unresponsive mutants have been isolated (*Pellegrini et al., 1989*). The U5A clone lacks a functional IFNAR2 receptor (*IFNAR2−/−*) and thus is impaired in type I interferon receptor signaling (*Lutfalla et al., 1995*). 2fTGH and U5A cells were both highly permissive to *Listeria* infection (*Figure 1—figure supplement 1*). Strikingly in the U5A cells (defective for type I interferon binding and signaling), as in 2fTGH cells, there is still a robust ISG15 response to *Listeria* infection (*Figure 1G*).

Since B18R treatment does not inhibit type III interferon signaling, ISG15 induction could arise via type III interferon receptor activation (*Bandi et al., 2010*). However, 2fTGH cells are unresponsive to type III interferon (*Zhou et al., 2007*). Therefore, the ISG15 protein induction that we observed is independent of both type I and type III interferon signaling. Taken together, our results show that ISG15 can be induced by *Listeria* in an interferon-independent manner in human nonphagocytic cells. We thus sought to determine how ISG15 was induced and what consequences ISG15 expression had on the cell and on infection.

### Cytosolic *Listeria* induces ISG15

To determine which signaling pathway was responsible for the *Listeria*-induced ISG15 transcript and to help identify the cellular compartment in which bacteria are sensed, we made use of *Listeria* strains that are impaired at different stages of infection. Incubation of cells with *L. innocua*, a non-pathogenic *Listeria* species which cannot invade cells, did not induce ISG15 induction, demonstrating that external pathogen recognition receptors were not involved (*Figure 2A*). We then used a strain of

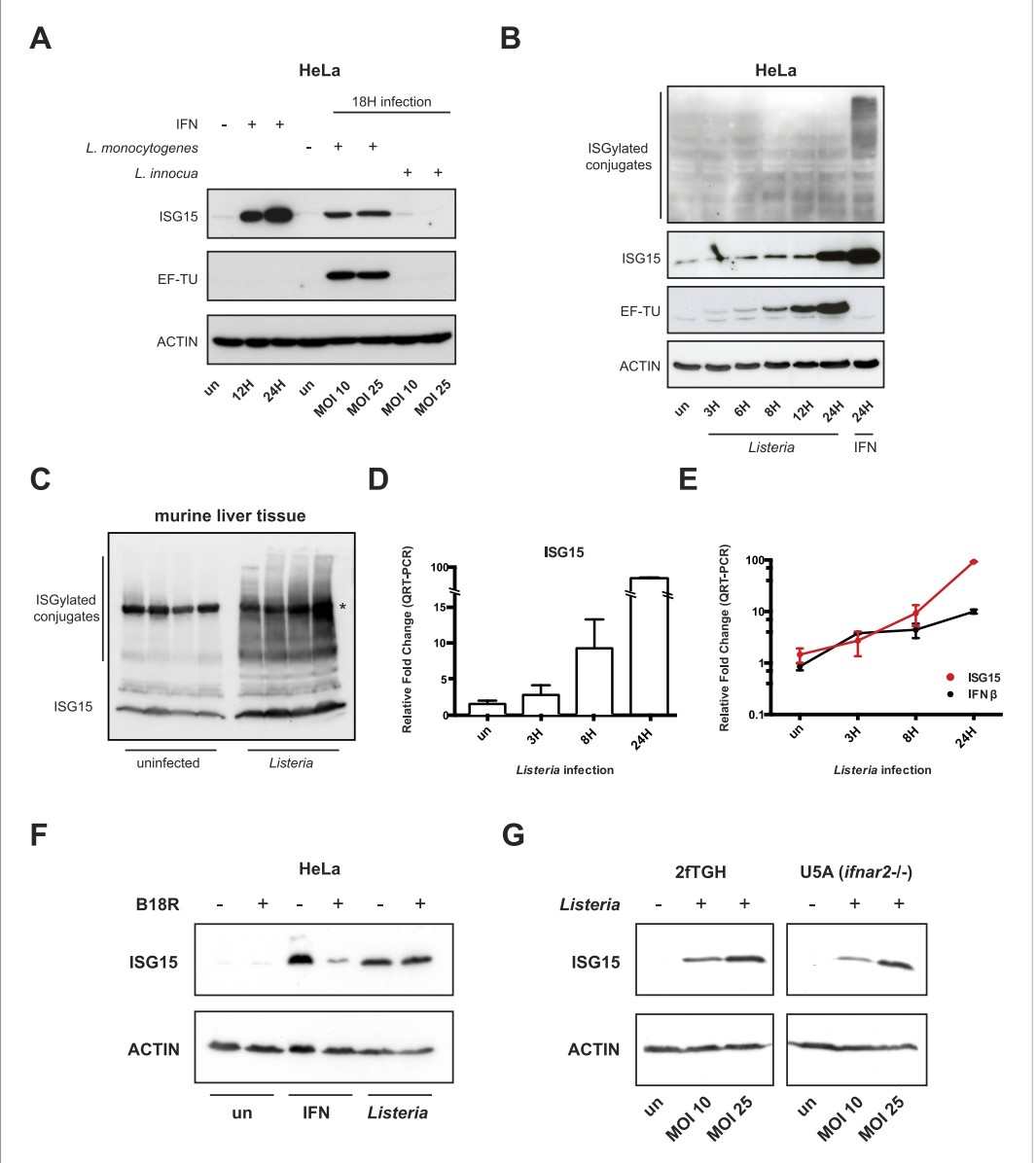

**Figure 1**. ISG15 is induced by *Listeria monocytogenes* infection both in vitro and in vivo and ISG15 induction can be type I interferon independent. (**A**) HeLa cells were lysed and immunoblotted with α-ISG15, α-EF-Tu (EF-Tu is a prokaryotic translation elongation factor that we use as an indicator of infection level), and α-ACTIN following 12 or 24 hr of interferon β treatment at 1000 units/ml, infection with *L. monocytogenes* for 18 hr at multiplicity of infection (MOI) of 10 or 25 bacteria per human cell, and incubation with *Listeria innocua* at MOI of 10 or 25 bacteria per human cell. (**B**) HeLa cells were lysed and immunoblotted with α-ISG15, α-EF-Tu, and α-ACTIN following a time course of *L. monocytogenes* infection from 3 to 24 hr, interferon β treatment for 24 hr was used as a positive control for ISGylation. (**C**) Liver tissue from mice injected with saline or infected with *Listeria* for 72 hr was lysed and immunoblotted with α-ISG15, each lane corresponds to a distinct animal (* indicates background band). (**D**) Relative fold change by qRT-PCR of ISG15 transcript following time course of infection with *Listeria*. (**E**) Relative fold change of ISG15 by qRT-PCR compared with the interferon β transcript over time course of infection with *Listeria*; data represented in a logarithmic scale. (**F**) Cells were lysed and immunoblotted with α-ISG15 and α-ACTIN following pre-treatment with the viral protein B18R followed by interferon α2 treatment (1000 μ/ml) or *Listeria* infection. (**G**) 2fTGH and U5A (*IFNAR2−/−*) cells were lysed and immunoblotted with α-ISG15 and α-ACTIN following *Listeria* infection for 18 hr.

The following figure supplement is available for figure 1:

**Figure supplement 1**. 2fTGH cells are permissive to *L. monocytogenes* infection.

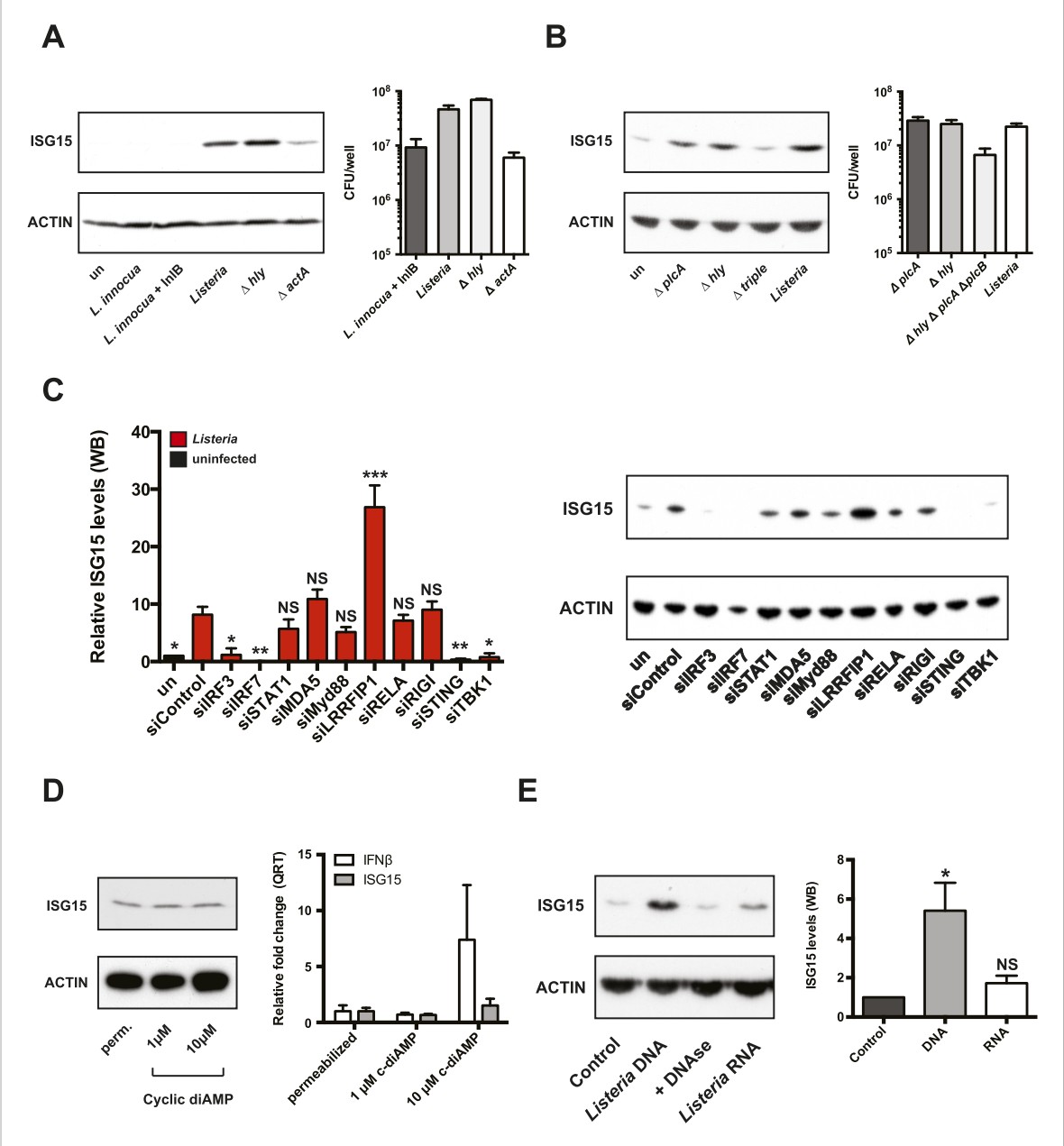

**Figure 2**. *Listeria* induces ISG15 via the cytosolic surveillance pathway (CSP) which senses bacterial DNA. (**A**) Cells were lysed and immunoblotted with the indicated antibodies following infection with various mutants of *Listeria*: *Listeria* strain EGD (MOI 10), *Δhly* (MOI 10), *ΔactA* (MOI of 50), and *L. innocua* + InlB (MOI of 100). Different MOIs were used at the outset in an attempt to equalize Colony forming units (CFUs) at the end of the experiment. CFUs per ml of intracellular bacteria following infection were determined by serial dilution after 18 hr of infection. (**B**) As before cells were lysed and immunoblotted with the indicated antibodies following infection with *Listeria* strain EGD-e PrfA* (MOI 10), *ΔplcA* (MOI 10), *Δhly* (MOI of 10), and a triple mutant of *ΔhlyΔplcAΔplcB* (MOI of 100). CFUs per ml of intracellular bacteria following infection were determined by serial dilution after 24 hr of infection. (**C**) HeLa cells were treated with siRNA pools targeting the indicated mRNA or siControl for 72 hr. Cells were then infected for 18 hr (MOI 10), lysed, and immunoblotted with α-ISG15 and α-ACTIN antibodies. Values were generated using ImageJ to quantify relative levels of induction of ISG15 compared to ACTIN. Values from three independent experiments are displayed (average ± SEM). siRNA knockdown was confirmed with qRT-PCR from a well of a technical replicate. Statistical significance calculated using ANOVA followed by Bonferonni's multiple comparison test against siControl. (**D**) Cells were lysed and immunoblotted with α-ISG15 and α-ACTIN following permeablization without Cyclic-diAMP, with 1 μM Cyclic-diAMP or with 10 μM Cyclic-diAMP. Average ± SEM; fold change of ISG15 or interferon β normalized to GAPDH levels following permeablization without Cyclic-diAMP, with 1 μM Cyclic-diAMP or with 10 μM Cyclic-diAMP. (**E**) Cells were lysed and immunoblotted with α-ISG15 and α-ACTIN following transfection with 800 ng of *Listeria* genomic DNA, *Listeria* genomic DNA treated with DNAase or total *Listeria* RNA for 24 hr; Average ± SEM; values were generated using ImageJ to quantify relative levels of induction of ISG15 compared to ACTIN in three independent experiments. Statistical significance calculated using ANOVA

*Figure 2. continued on next page*

*Figure 2. Continued*

followed by Bonferonni's multiple comparison test against transfection control. Statistical significance is indicated as follows: NS, nonsignificant; *p < 0.05; **p < 0.01; ***p < 0.001.

The following figure supplements are available for figure 2:

**Figure supplement 1**. *Listeria ΔhlyΔplcAΔplcB* does not escape the vacuole.

**Figure supplement 2**. HeLa cells express *STING* mRNA.

*L. innocua* that expresses Internalin B (InlB), a *L. monocytogenes* virulence factor that mediates entry into nonphagocytic cells (*Dramsi et al., 1995*). These bacteria enter the cell and are entrapped in a membrane-bound phagosome, but lack the required virulence factors to escape from it. This strain was also unable to induce an ISG15 signal, suggesting that the pathogen recognition receptors that survey the phagosome are not sufficient for ISG15 induction (*Figure 2A*). *Listeria's* hemolysin, listeriolysin O (LLO) is an extremely potent virulence factor, which triggers vacuolar escape of the bacterium as well as a plethora of changes in the host cell (*Hamon et al., 2012*). Strikingly, we found that the Δ*hly* mutant was able to potently induce ISG15 (*Figure 2A*). Thus, LLO is not necessary for ISG15 induction. However, in several human epithelial cell lines the mutant that lacks LLO (Δhly) can still escape into the cytosol (*Portnoy et al., 1988*; *Marquis et al., 1995*). *Listeria* expresses two phospholipases that can compensate for the lack of LLO in the Δ*hly* mutant in order to free the bacterium from the phagosome in human epithelial cells (*Marquis et al., 1995*; *Smith et al., 1995*). To assess whether *Listeria* trapped in the phagosome could induce ISG15, we constructed a triple mutant (lacking LLO, PLCA, and PLCB) of *Listeria*. This mutant is unable to escape the phagosome of human epithelial cells (*Figure 2—figure supplement 1*). Single mutants (either in PLCA or LLO), which escape into the cytosol, induce a strong ISG15 signal relative to non-infected cells (*Figure 2B*). In contrast, the triple mutant that is confined to the phagosome does not induce ISG15 (*Figure 2B*). We thus conclude that only cytoplasmic bacteria induce ISG15. In fact, the only other mutant to induce less ISG15 production was the Δ*actA* mutant (*Figure 2A*). This mutant is unable to spread from cell to cell and cannot escape autophagic recognition, degradation, and lysis (*Gouin et al., 2005*; *Yoshikawa et al., 2009*). As a result bacterial load is much lower compared to wild-type bacteria, providing an explanation for the reduced ISG15 signal (*Figure 2A*). Taken together, our results reveal that ISG15 induction stems from cytosolic bacteria.

## ISG15 is induced via the CSP following sensing of cytosolic DNA

In order to determine which pathway was essential for ISG15 induction, we performed an siRNA screen of innate immune molecules that are known to be involved in bacterial sensing (*Figure 2C*). As for the experiments described above, we used HeLa cells for the siRNA screen. Although HeLa cells are reported to lack STING (*Burdette and Vance, 2013*), the ATCC line we worked with expressed *STING* mRNA, as evidenced by qRT-PCR. We were able to specifically extinguish this signal with siRNA (*Figure 2—figure supplement 2*). Our data showed that the ISG15 signal was clearly dependent on IRF3, IRF7, STING, and TBK1, implicating the CSP. In further support of an interferon-independent signal, depleting *STAT1*, which is a critical mediator of type I and III interferon signaling did not abrogate the ISG15 signal (*Figure 2C*). In non-immune cells interferon induction has been linked to sensing of triphosphorylated RNA by RIG-I (*Abdullah et al., 2012*; *Hagmann et al., 2013*); however, in our experimental conditions, RIG-I did not seem to be required for the ISG15 signal. Instead, it seems that direct ISG15 induction occurs through a pathway similar to the CSP in macrophages.

We next sought which PAMP was necessary and sufficient for ISG15 induction. We transfected cells with either *Listeria* genomic DNA, *Listeria* genomic DNA pre-treated with DNAse, *Listeria* total RNA, or we permeabilized cells in the presence of cyclic-di-AMP. To control whether the cyclic di-AMP was biologically active and reached the cytosol of the cells, we assessed *IFNB1* levels by qRT-PCR (*Figure 2D*). As expected *Listeria* cyclic di-AMP led to an increase in *IFNB1* transcript levels. Although *IFNB1* induction with cyclic di-AMP was lower than that reported for murine macrophages

(*Woodward et al., 2010*), it was similar to that reported for human phagocytes and monocytes (*Hansen et al., 2014*). *Listeria* genomic DNA was the only PAMP sufficient for an increase in ISG15 levels (*Figure 2E*), whereas *ISG15* levels following cytosolic exposure to cyclic-di-AMP did not increase (*Figure 2D*). Collectively, these data implicate that the CSP can directly induce ISG15 after sensing of bacterial DNA in the cytosol of cells in a pathway that requires STING, TBK1, IRF3, and IRF7.

## ISG15 protects against *Listeria* infection in vitro and in vivo

We next assessed whether ISG15 has a functional effect on infection. We created a retroviral construct that expresses an epitope-tagged version of ISG15 (3XFlag-6His-ISG15). We then infected cells that stably express 3XFlag-6His-ISG15 with *Listeria*. As a control, cells were retrovirally transduced with pBabe puro empty vector. For the same multiplicity of infection (MOI) after 3 hr of infection, stable overexpression of ISG15 resulted in 50% fewer cytosolic bacteria as compared to control cells (*Figure 3A*). We then assessed bacterial uptake by differentiating between the bacteria that are inside the cell or those that remain on the surface (inside-out staining) and did not observe a difference between control and ISG15-overexpressing cells for invasion (*Figure 3—figure supplement 1A*). During a time course of infection in these cells at 7 and 12 hr, there were still 50% fewer bacteria, and by 24 hr, the levels of bacteria had equalized between the two cell lines (*Figure 3—figure supplement 1B*). This time course suggests that ISG15 does not impact bacterial replication as an active bacterial clearance mechanism would (*Figure 3—figure supplement 1B*). We then knocked down *ISG15* during infection. This increased bacterial load by nearly twofold after 15 hr (*Figure 3B*, *Figure 3—figure supplement 2*). These data strongly suggest that ISG15 plays a role in protection against *Listeria* infection following uptake. To further explore this phenotype in primary cells, we isolated mouse embryonic fibroblasts (MEFs) from wild-type and *Isg15*−/− embryos. We infected these cells with *Listeria* and we observed a fivefold increase in bacterial load in *Isg15*−/− MEFs compared to wild-type MEFs for the same MOI (*Figure 3C,D*). Interestingly, *Isg15*−/− MEFs are not susceptible to other intracellular pathogens such as *Shigella flexneri* and *Salmonella typhimurium*, and in HeLa cells only *Staphylococcus aureus* is able to induce as much ISG15 as *Listeria* (*Figure 3—figure supplement 1C,D*). Indeed, *S. flexneri* and *S. typhimurium* induce very little ISG15. We next examined whether ISG15 could also play a role during *Listeria* infection in vivo by assessing the susceptibility of *Isg15*−/− animals to the pathogen during systemic infection (*Osiak et al., 2005*). ISG15-deficient mice exhibited a significant increase in bacterial load compared to wild-type animals in both the spleen and liver after 72 hr of systemic sub-lethal *Listeria* infection (*Figure 3E,F*). Taken together, these data demonstrate that ISG15 restricts *Listeria* infection both in vitro and in vivo.

## ISGylation machinery is induced by *Listeria* and ISGylation protects against infection

Since ISG15 can mediate its protective effect through conjugation-dependent or conjugation-independent mechanisms, we wanted to determine if conjugation contributed to ISG15's role in defense against *Listeria*. Therefore, we initially assessed whether *Listeria* could induce the enzymes required for ISGylation. To this end, we performed RNASeq of Lovo cells infected with *Listeria* for 24 hr compared to uninfected cells. The RNASeq data has been uploaded to Array Express with the accession E-MTAB-3649 (*Radoshevich et al., 2015b*). We found that *UBE1L* (E1), *UBE2L6* (E2), *HERC5* and *TRIM25* (E3s) mRNA are all significantly upregulated following *Listeria* infection, as is the deconjugating enzyme *USP18* (*Figure 3G*). In order to assess whether UBE2L6 and TRIM25 are induced as rapidly as ISG15 at the protein level, we monitored their expression by immunoblot at 3 hr post infection (*Figure 3H*). Interestingly, TRIM25 already displayed increased expression relative to loading control at 3 hr post infection. UBE2L6 was present in these cells at this time point as well. We then sought to determine whether conjugation-incompetent MEFs would have a higher bacterial load following *Listeria* infection. Primary MEFs that lack UBE1L, and thus can not form ISG15 conjugates, have a much higher bacterial burden following *Listeria* infection than wild-type MEFs at 4 hr post infection. This phenotype mirrors the bacterial burden of *Isg15*−/− MEFs, clearly indicating that ISG15's role in host defense against *Listeria* in vitro requires ISGylation (*Figure 3I*).

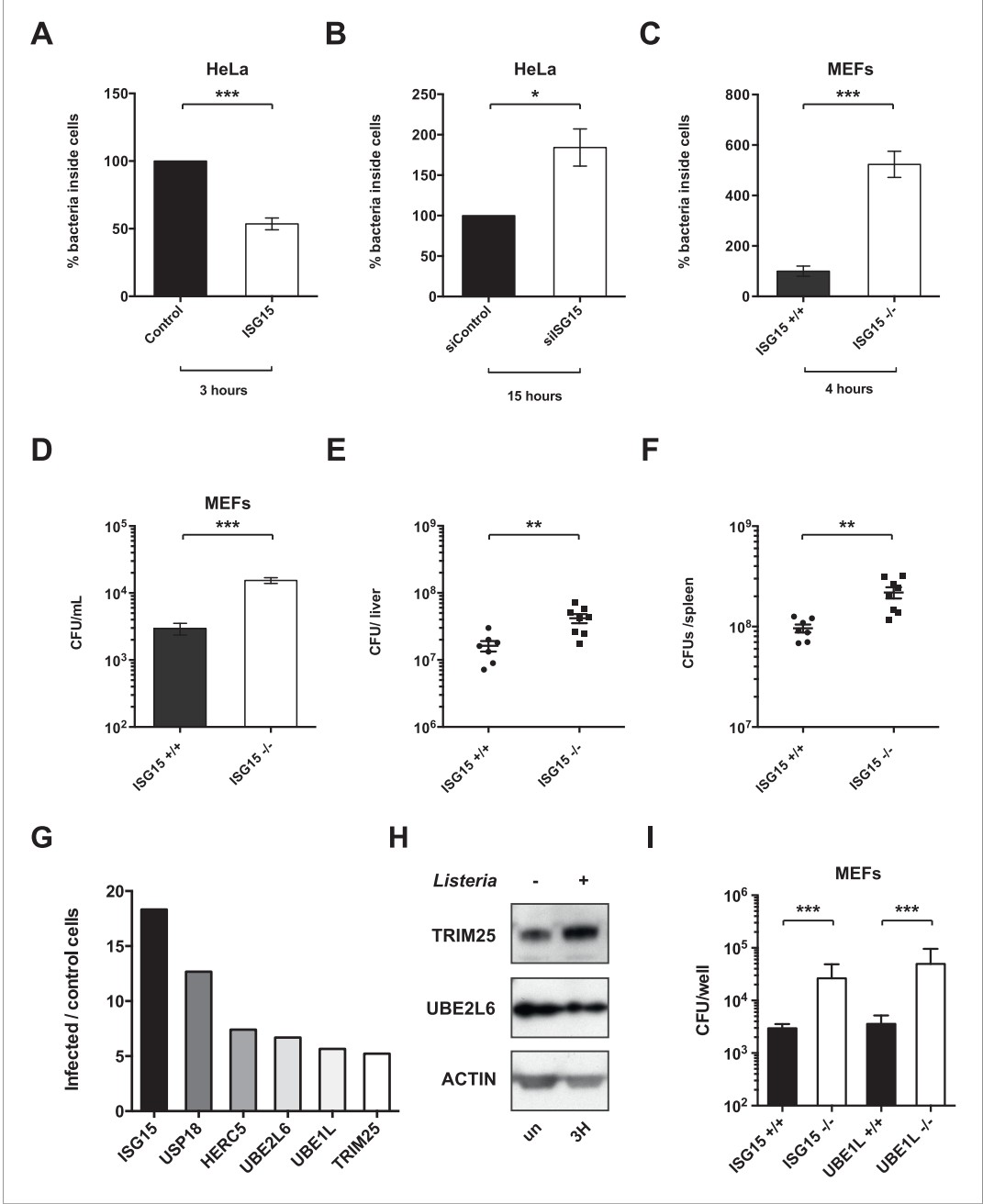

**Figure 3**. ISG15 protects against *Listeria* infection in vitro and in vivo. (**A**) Percentage of bacteria inside HeLa cells, infected at an MOI of 25, which were transduced with empty vector (Control) or stably express ISG15 after 3 hr of infection; CFUs of bacteria within control cells were normalized to 100%, data shown is AVG ± SEM. Statistical significance were determined using two tailed t-test. (**B**) Percentage of bacteria inside HeLa cells which have been transfected with siControl or siISG15 (15 hr of infection, MOI 25). siControl cells were normalized to 100%, data is shown as AVG ± SEM. Statistical significance determined using two tailed t-test. (**C**) Primary mouse embryonic fibroblasts (MEFs, *Isg15*+/+ or *Isg15*–/–) were infected with *Listeria* for 4 hr at an MOI of 10. CFUs of bacteria within *Isg15*+/+ cells were normalized to 100%, and error bars represent ± SEM. Statistical significance determined using two tailed t-test. (**D**) Data shown is average CFUs per ml ± SEM in *Isg15*+/+ MEFs vs *Isg15*–/– MEFs following 4 hr of infection, MOI 10. (**E**) and (**F**) *Isg15*+/+ or *Isg15*–/– mice were infected intravenously with 5 × 10^5 of *Listeria* strain EGD. The liver and spleen were isolated following 72 hr of infection, and CFUs per organ were calculated by serial dilution and replating; circles or squares depict individual animals. The line denotes AVG ± SEM. Significance for in vivo data determined using Mann–Whitney test. (**G**) RNASeq data of significantly upregulated ISG15-related genes compared with non-infected controls in LoVo cells following *Listeria* infection for 24 hr. (**H**) Cells were lysed and
*Figure 3. continued on next page*

*Figure 3. Continued*

immunoblotted with the indicated antibodies following infection with *Listeria* for 3 hr. (**I**) Data shown is average CFUs per ml ±SEM in *Isg15*+/+ MEFs vs *Isg15*−/− MEFs and in *Ube1L*+/+ and *Ube1L*−/− MEFs following 4 hr of infection, using an MOI of 10. Statistical significance is indicated as follows: NS, nonsignificant; *p < 0.05; **p < 0.01; ***p < 0.001.

The following figure supplements are available for figure 3:

**Figure supplement 1**. ISG15 protects cells from *Listeria* infection at early time points.

**Figure supplement 2**. siRNA-mediated knockdown effectively depletes ISG15 during *Listeria* infection.

## ISGylation modifies ER and Golgi proteins and increases canonical secretion

Since ISG15 is a ubiquitin-like modifier that is known to be covalently linked to hundreds of cellular and several viral substrates (*Giannakopoulos et al., 2005*; *Zhao et al., 2005*), our goal was to identify which substrates following overexpression of ISG15 could account for the protective effect detected in the context of *Listeria* infection and used a proteomic approach. We made use of SILAC coupled with LC-MS/MS and compared cells that express empty vector, cells that express ISG15 and cells that express ISG15 that were treated with interferon, the primary inducer of ISGylation (*Figure 4A,B*). We identified thirty ISGylated proteins modified following overexpression of ISG15 (*Figure 4C–E*, *Figure 4—source data 1*, depicted in blue). The proteomics data have been deposited to the ProteomeXchange Consortium (*Vizcaino et al., 2014*) via the PRIDE partner repository with the data set identifier PXD001805 (*Radoshevich et al., 2015a*). Interestingly, these proteins have not yet been reported to be targets of ISGylation. Following interferon treatment, ISG15 modified twelve additional proteins distinct from those that were ISGylated after overexpression without treatment (*Figure 4C–E*, *Figure 4—source data 1*, depicted in red). Three of these proteins are known targets of ISGylation following interferon treatment in the aforementioned screens. The other nine are novel substrates of ISGylation. To gain more insight into the role of ISGylation of the modified substrates following overexpression, we performed a gene ontology (GO) analysis of the thirty ISGylated proteins. To our surprise over 80% of ISG15-target proteins are integral membrane proteins (*Figure 5A*). Even more intriguingly they are known to be primarily localized to the endoplasmic reticulum and Golgi apparatus and/or are critical for glycosylation, ER morphology and ER to Golgi trafficking (e.g., the oligosacchary-transferase (OST) complex, RTN4, ATL3, SEC22B, ERGIC1, and ERGIC3; *Figure 4C*). One of the proteins enriched following ISG15 overexpression is Magnesium Transporter 1 (MAGT1). MAGT1 is critical for T cell activation and patients with a deletion in the *MAGT1* gene are susceptible to viral and certain bacterial infections (*Li et al., 2014*). The protein is localized to the cell surface or to the ER where it interacts with the OST complex, which is critical for N-glycosylation (*Pfeffer et al., 2014*). Notably, we also identified multiple proteins from the OST complex as targets of ISGylation (*Figure 4C*, inset). To validate that MAGT1 was modified by ISG15, we affinity purified ISGylated proteins and performed immunoblot analysis (*Figure 5B*). Two higher molecular weight ISG15-MAGT1 complexes appear that are absent in control cells (*Figure 5B*, arrows). To assess whether MAGT1 was ISGylated following infection, we immunoprecipitated ISG15 and could show that there is a higher molecular weight MAGT1 complex that immunoprecipitates with ISG15 (*Figure 5—figure supplement 1A*). Additionally, non-conjugated MAGT1 was enriched following ISG15 immunoprecipitation as well. Whether the interaction is direct or indirect within a multi-protein complex remains to be determined. Of note, MAGT1 is also induced by *Listeria* infection (*Figure 5—figure supplement 1A*). A second target of ISGylation following ISG15 overexpression is Reticulon 4 (RTN4) (*Figure 4C,D*). It is known that RTN4 helps to determine the morphology of the endoplasmic reticulum (*Voeltz et al., 2006*). We also validated that endogenous RTN4 was modified by ISG15 by immunoblot (*Figure 5B*), and we detected an upshifted ladder of modified RTN4 that was also recognized by the ISG15 antibody. ISG15 is not known to make chains so the ladder most likely corresponds to either multiple ISGylations of RTN4 or concomitant modifications of RTN4 by ISG15 together with other modifiers as opposed to a single modification.

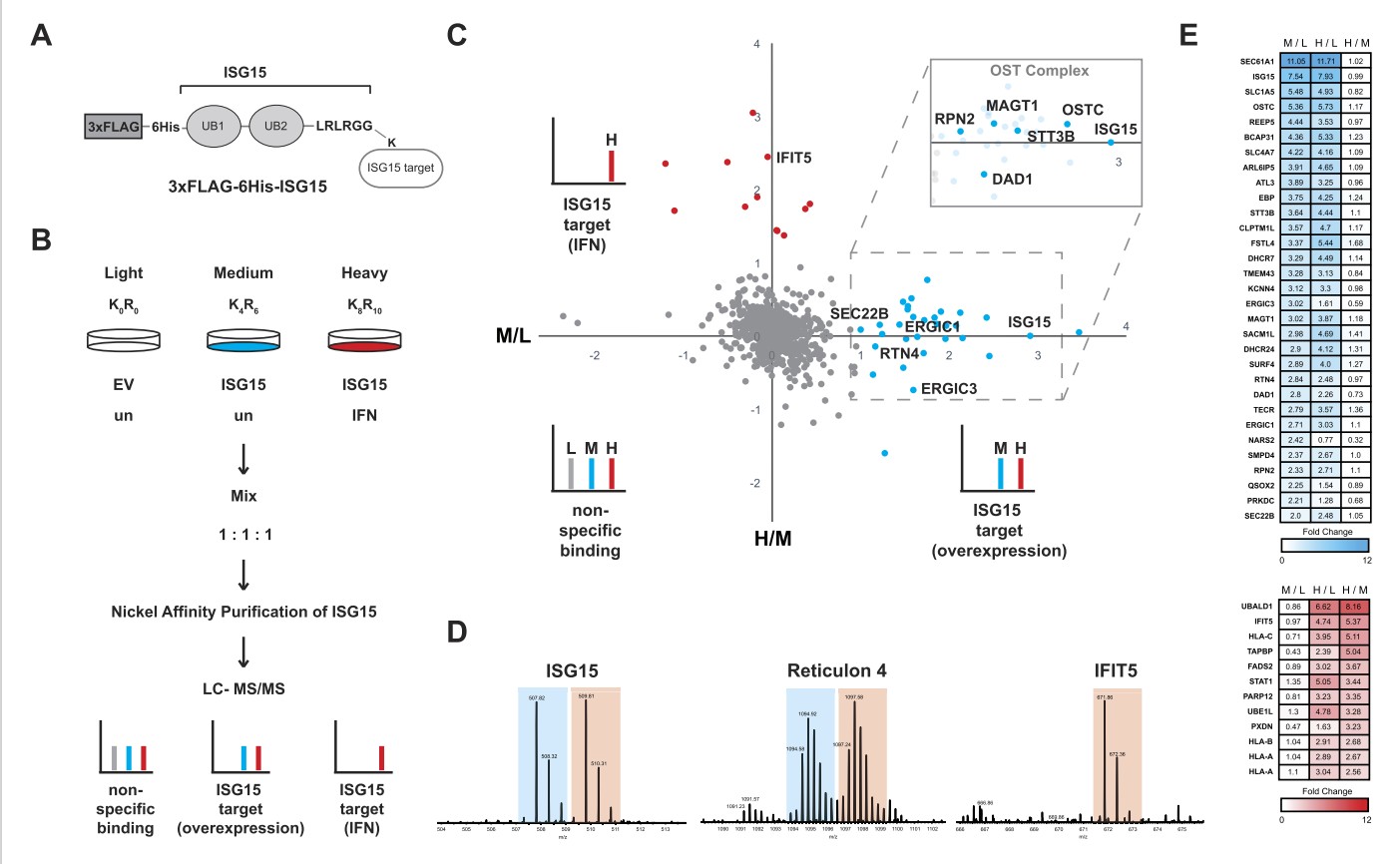

**Figure 4**. Schematic representation of ISG15 targets identified by proteomics. (**A**) Epitope-tagged ISG15 (3XFLAG-6His-ISG15) represented binding to an unknown target via isopeptide linkage. ISG15 has a stop codon engineered following the LRLRGG motif truncating the amino acid sequence which in wild-type ISG15 would be removed by USP18 during maturation. (**B**) Scheme for enrichment of ISGylated targets using SILAC. Equal protein amounts of each condition were combined, nickel affinity purified, and then analyzed by LC-MS/MS. (**C**) Scatter plot of $Log_2$ transformed ISGylated proteins. The X axis displays the medium (M) condition over light (L), which is ISG15 overexpression over empty vector control (in blue), and the Y axis displays the heavy (H) condition over medium (M), which is interferon treatment over ISG15 expression (in red). Cut-off was determined as M/L ratio of greater than or equal to one for ISG15 targets and H/M ratio of greater than or equal to one for interferon ISGylation targets. Inset is a representation of ISG15 targets from the OST Complex. Other targets identified are proteins involved in ER morphology or ER to Golgi transport. (**D**) Representative MS spectra from ISG15, RTN4 and IFIT5 peptides: $_{100}$LTQTVAHLK$_{108}$ from ISG15, 2+; $_{58}$KPAAGLSAAPVPTAPAAGAPLMDFGNDFVPPAPR$_{91}$ from RTN4, 3+ and $_{207}$AVTLNPDNSYIK$_{218}$ from IFIT5, 2+. (**E**) Heat map of fold change of ISGylated proteins following ISG15 overexpression (in blue) and of ISGylated proteins following interferon treatment (in red).

The following source data is available for figure 4:

**Source data 1**. Enlarged heat map with protein identifiers related to *Figure 4E*.

Here again we could validate the endogenous modification of RTN4 by ISG15 following *Listeria* infection (*Figure 5—figure supplement 1A*) by detecting multiple distinct slower migrating bands of RTN4 and ISG15 (*Figure 5—figure supplement 1A*). Interestingly, Atlastin-3, another protein that affects ER morphology was also identified as an ISG15 target. We attempted to validate this interaction by immunoblot but were unable to identify an upshifted complex, in part due to inadequate antibody sensitivity (data not shown). Atlastin-3 is an ER dynamin-like GTPase that can alter the morphology of both the ER and the Golgi apparatus (*Rismanchi et al., 2008*; *Hu et al., 2009*). RTN4 and Atlastin-3 can interact to help shape the tubular ER. When RTN4 is overexpressed it increases protein disulphide-isomerase (PDI) clustering in the cell (*Yang et al., 2009*; *Bernardoni et al., 2013*). We thus assessed PDI puncta formation following ISG15 overexpression and in line with our hypothesis found a dramatic increase in puncta formed upon ISG15 expression (*Figure 5C*).

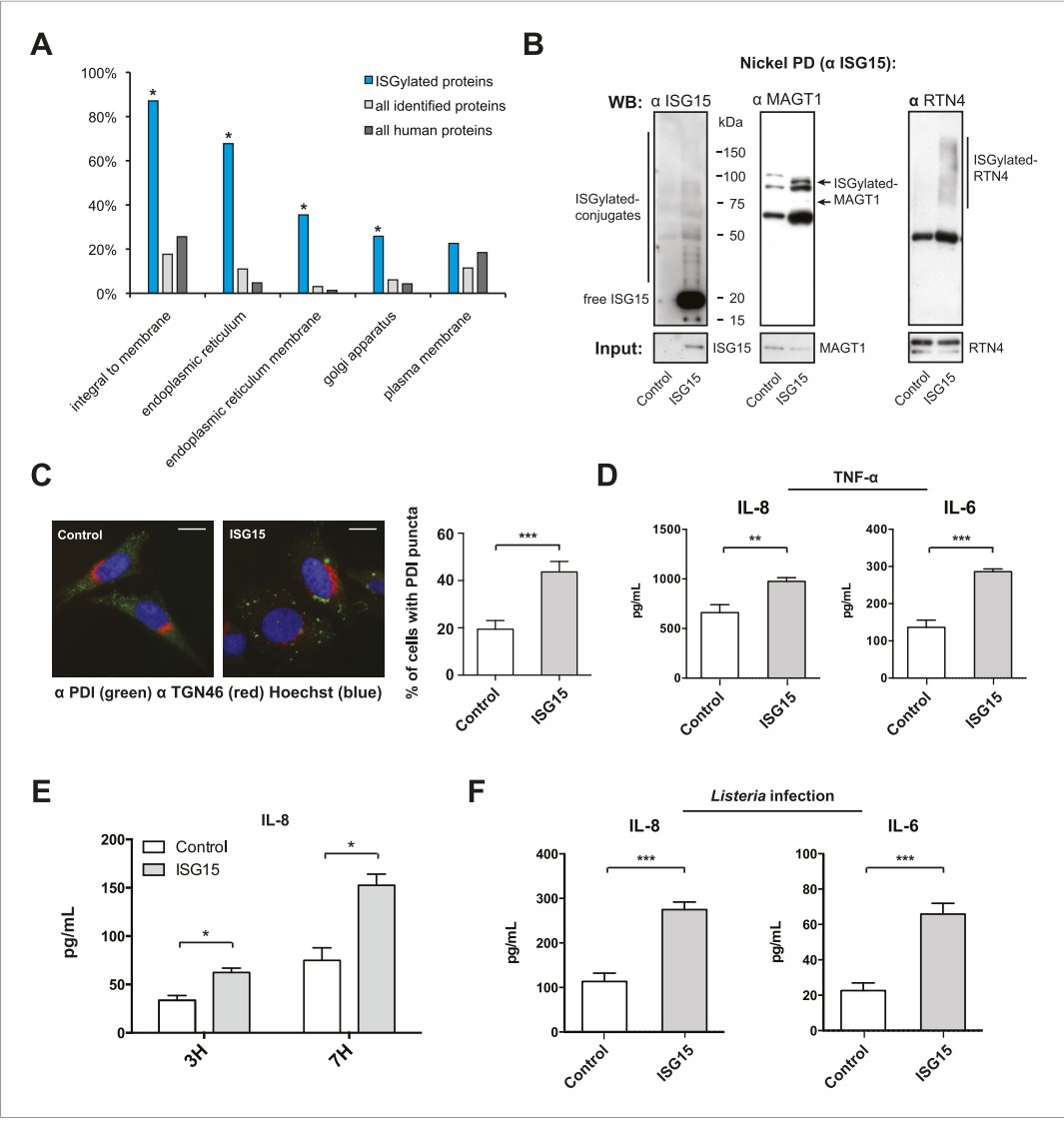

**Figure 5**. ISGylation modifies ER and Golgi proteins and increases canonical secretion. (**A**) Gene ontology (GO) analysis of ISGylated proteins enriched following ISG15 overexpression (blue) relative to all proteins identified in this analysis (light gray) and all human proteins as annotated by the Uniprot/Swiss-prot database (dark gray). (**B**) Nickel affinity purification of ISG15 in empty-vector control or ISG15 overexpression cells followed by α-ISG15, α-MAGT1, and α-RTN4 immunoblot. Input is 0.1% of the initial affinity purification volume. Higher molecular weight species corresponding to ISGylated RTN4 or MAGT1 are indicated. (**C**) Cells were fixed and immunostained with α-PDI antibody (green), with α-TGN46 (red) and Hoechst (blue). Cells with four or more large puncta were enumerated as cells with PDI clusters, a minimum of 100 cells were counted per experiment, and data are shown as percentage of cells with PDI clusters; AVG ± SEM. Statistical significance determined using two-tailed t-test. (**D**) The indicated cells were treated with 20 ng/ml TNF-α for 24 hr in a 96-well format. 100 μl of supernatant was collected for each ELISA and assayed for the presence of IL-8 and IL-6. Statistical significance determined using two-tailed t-test. (**E**) The indicated cells were infected for 3 or 7 hr with *Listeria* (MOI 25) in a 96-well format, supernatants were collected and 100 μl of supernatant was used for an IL-8 ELISA. Statistical significance determined using two-tailed t-test. (**F**) The indicated cells were either infected with *Listeria* (MOI 25) for 24 hr in a 96-well format. 100 μl of supernatant was collected for each ELISA and assayed for the presence of IL-6 or IL-8. Statistical significance determined using two-tailed t-test. Statistical significance is indicated as follows: NS, nonsignificant; *p < 0.05; **p < 0.01; ***p < 0.001.

The following figure supplements are available for figure 5:

**Figure supplement 1**. ISG15 covalently modifies RTN4 and MAGT1 following infection.

*Figure 5. continued on next page*

*Figure 5. Continued*

**Figure supplement 2**. Non-biased assay of 31 cytokines following TNFα treatment in control vs ISG15-overexpressing cells.

**Figure supplement 3**. ISG15 is not secreted as a cytokine from HeLa cells following TNFα treatment or *Listeria* infection.

We further assessed the ER morphology of ISG15-overexpressing cells by visualizing SEC61-GFP or KDEL-BFP. Neither of these markers displayed an apparent redistribution in ISG15 overexpressing cells relative to control cells (*Figure 5—figure supplement 1B*). We also assayed whether there was a difference in ER mass using ER-tracker and FACS analysis (*Figure 5—figure supplement 1C*). There was no detectable difference between the two cell lines in ER mean fluorescence intensity. Taken together, the most striking effect of ISG15 overexpression on ER morphology is PDI clustering and distribution.

Certain UBL proteins, such as SUMO and ISG15, modify 1–5% of a given substrate; nevertheless, this partial modification is sufficient to lead to a phenotypic effect on the substrate. We thus investigated whether the enrichment of ER and Golgi proteins modified by ISG15 could have an effect on the primary function of these organelles. Canonical secretion of cytokines and growth factors involves translation and import of the proteins into the ER, folding and modification (e.g., glycosylation), within the ER, followed by sorting and trafficking within the Golgi culminating in targeting to the plasma membrane. Since we found proteins implicated in many of these processes to be ISGylated (*Figure 4C*), we thus hypothesized that canonical secretion could be altered in the ISG15-overexpressing cells. As TNF-α is known to lead to canonical secretion of many cytokines, via activation of NF-κB and MAPK pathways, we assessed secreted cytokine levels in ISG15-overexpressing cells compared to control cells following TNF-α treatment. Among a panel of 31 human cytokines, we detected increased secretion of IL-8 and IL-6 in ISG15-expressing cells relative to control cells (*Figure 5—figure supplement 2*). We validated these findings with quantitative ELISAs for IL-6 and IL-8 (*Figure 5D*). ISG15-expressing cells secreted significantly more IL-6 and IL-8 following TNF-α treatment than control cells (*Figure 5D*). As ISG15 itself has been reported to be able to act as a secreted cytokine, we also assessed ISG15 in the supernatant. However, we did not observe secreted ISG15 from either cell type (*Figure 5—figure supplement 3*).

Finally, based on the results observed following TNF-α treatment, it was of critical importance to test whether a similar ISG15-dependent modulation of cytokine secretion occurred during *Listeria* infection. As shown in *Figure 5E*, following infection, ISG15-overexpressing cells secreted significantly more IL-8 than infected wild-type cells at both 3 and 7 hr post infection. The difference in IL-8 concentration in the media after 24 hr of infection was nearly threefold higher in ISG15-overexpressing cells relative to control cells (*Figure 5F*). As with IL-8 the IL-6 concentration in the media increased by threefold relative to control cells following 24 hr of infection. Moreover, we could not detect ISG15 in the supernatant following infection in either cell line (*Figure 5—figure supplement 3*). In order to confirm this effect on cytokine secretion under endogenous conditions, we assayed whether ISG15 could modulate cytokine secretion following infection with *L. monocytogenes* as opposed to *L. innocua* expressing InlB. Infection with *L. monocytogenes* led to significantly higher IL-8 and IL-6 than *L. innocua* expressing InlB, which correlates with increased ISG15 expression (*Figure 5—figure supplement 1D*). Furthermore, we assessed cytokine secretion in transformed MEFs that lack ISG15, relative to their wild type counterparts following *Listeria* infection. Despite significantly higher levels of bacteria in the *Isg15*−/− MEFs, the cells secreted significantly less IL-6 than wild-type cells (*Figure 5—figure supplement 1E*). Therefore, ISG15 appears to be a critical modulator of cytokine secretion following *Listeria* infection. Collectively, our data reveal that during *Listeria* infection ISG15 modification of distinct ER and Golgi proteins increases secretion of cytokines which are known to counteract infection in vivo.

## Discussion

ISG15 has primarily been characterized, in the context of infection, as an antiviral protein induced by type I interferon. Here, we show for the first time, to our knowledge, that *Listeria* infection can lead to

the direct induction of an interferon-stimulated gene, ISG15, in a type I interferon-independent manner in nonphagocytic cells. This induction requires STING, TBK1, IRF3 and IRF7 and *Listeria* DNA is necessary and sufficient to induce the signal required to produce ISG15. We have demonstrated that ISG15 restricts infection of nonphagocytic cells. Furthermore, primary fibroblasts that lack ISG15 or UBE1L are more susceptible to infection and animals deficient in ISG15 have a higher bacterial burden both in the liver and spleen. Finally, our proteomic analysis has helped to reveal a novel mechanism through which ISG15 and ISGylation can restrict *Listeria* infection. Furthermore, we demonstrated a previously unidentified ISGylation of integral ER and Golgi proteins, which in turn correlates with increased cytokine secretion upon infection.

## Interferon-independent induction of ISG15 during *Listeria* infection

*Listeria* has long been used as a model organism to understand the adaptive immune response, however more recently interest in the field has focused on the innate immune pathways required to produce type I and type III interferons and bacterial modulators of type I and type III interferons during *Listeria* infection of nonphagocytic cells (*Lebreton et al., 2011*; *Abdullah et al., 2012*; *Bierne et al., 2012*; *Hagmann et al., 2013*; *Odendall et al., 2014*). In phagocytic cells, the primary type I interferon response to *Listeria* is mediated by *Listeria* cyclic-di-AMP (*Woodward et al., 2010*). In nonphagocytic cells, type I interferon is produced downstream of RIG-I sensing of triphosphorylated RNA molecules (*Abdullah et al., 2012*; *Hagmann et al., 2013*). Here, we highlight a complementary pathway, which can lead to rapid and direct interferon-independent induction of ISG15, and potentially other ISGs, emanating from sensing of bacterial DNA. Notably, we implicate the same molecular players required for the CSP in macrophages. However cyclic-di-AMP, the PAMP sensed by the CSP in mouse macrophages, was not sufficient to directly induce ISG15 in an interferon-independent manner in nonphagocytic cells. Interestingly, a similar pathway was implicated in sensing the cytosolic bacterial DNA of *Mycobacterium tuberculosis* in macrophages (*Manzanillo et al., 2012*) and sensing viral DNA in nonphagocytic cells (*Hasan et al., 2013*). Our study complements and supports these findings as it also alludes to the importance of this response in nonphagocytic cells during *Listeria* infection.

## ISG15 as a novel modulator of canonical secretion

It is still an open question whether specific ISGs can contribute to a functional antibacterial state similar to an antiviral state. We hypothesized that the massive induction of ISG15 following *Listeria* infection must have a phenotypic effect on host cells. Indeed, we have demonstrated that ISG15 counteracts *Listeria* infection both in vitro and in vivo. The in vitro susceptibility is clearly conjugation-dependent and we have preliminary data suggesting that the conjugation incompetent mutant of ISG15 does not protect cells from infection (data not shown). During viral infections ISG15 can restrict infection by direct modification of viral proteins (*Zhao et al., 2010*), by modification of host proteins that are important to viral replication (*Pincetic et al., 2010*), by non-specific modification of newly translated proteins to target viral capsid (*Durfee et al., 2010*) and finally in the context of infection in vivo it can have non-conjugation dependent roles either within the cell or as a cytokine when secreted from polymorphonuclear leukocytes (*D'Cunha et al., 1996*; *Werneke et al., 2011*; *Bogunovic et al., 2012*). Here, we highlight a novel mode of action of ISG15 on canonical cytokine secretion that correlates with modification of ER and Golgi proteins. Interestingly, ubiquitin itself can have an effect on secretion by playing a role in determining COPII vesicle size and cargo through specific Cullin E3 ligases (*Lu and Pfeffer, 2014*). To our knowledge, a link between the intracellular function of ISG15 and canonical secretion of other cytokines has not been reported, thus our work identifies a novel function of ISG15, which could also play a role in other bacterial and viral infections.

Two recent studies have underscored the importance of ISG15 in the human innate immune response to pathogens by characterizing patients who lack a functional ISG15 protein (*Bogunovic et al., 2012*; *Zhang et al., 2014*). ISG15-deficient patients were susceptible to *Mycobacterium bovis* Bacille Calmette-Guérin (BCG). These individuals were shown to have reduced production of interferon γ, which was linked to ISG15's extracellular role as a cytokine. In light of our results, ISG15's intracellular role as a modulator of cytokine secretion in nonphagocytic cells could also be responsible for reduced production of interferon γ.

### Role of cytokine secretion during *Listeria* infection in vivo

Increased cytokine secretion mediated by ISG15 is a previously uncharacterized way for the host to engage the immune system to combat both bacterial and viral infection; the first cells that encounter *Listeria*, (i.e., nonphagocytic cells) which are able to induce ISG15 at early time points could raise the alarm against an invading pathogen and as a result help alert innate immune cells to the site of infection. In particular, IL-6 and IL-8 are key mediators of neutrophil and monocyte recruitment in vivo and IL-6 has been repeatedly characterized as a critical mediator of the innate immune response to *Listeria* (*Baggiolini and Clark-Lewis, 1992*; *Dalrymple et al., 1995*; *Personnic et al., 2010*). If cytokine secretion in the absence of ISG15 is less efficient, it could help to explain the moderate but statistically significant susceptibility of *Isg15*–/– mice to systemic *Listeria* infection. These data coupled with the aforementioned human ISG15-deficient patient susceptibility to BCG infection highlight the in vivo relevance of ISG15 as an antibacterial factor (*Bogunovic et al., 2012*). Finally, the interferon-independent induction of ISG15 and potentially other ISGs may help to explain the still perplexing phenotype of interferon receptor-deficient mice (*Ifnar*–/–). IFNAR-deficient mice are highly resistant to *Listeria* infection when infected intravenously (*Auerbuch et al., 2004*; *Carrero et al., 2004*; *O'Connell et al., 2004*; *Stockinger et al., 2004*). If certain subsets of cells are in fact able to induce ISGs in the absence of interferon and if these ISGs like ISG15 could contribute to an antibacterial state, this may explain why these mice are resistant to *Listeria* infection.

## Materials and methods

### Materials

Dr Sandra Pellegrini (Institut Pasteur) kindly provided the 2fTGH and U5A (*IFNAR2*–/–) cell lines along with an α-ISG15 antibody (gift from A Haas). Dr Matthew Albert and Dr Scott Werneke (Institut Pasteur) generously provided the *Ube1L*–/– MEFs. Dr Gia Voeltz (University of Colorado) generously provided SEC61-GFP and KDEL-BFP constructs. Dr Sabrina Jabs generously provided a plasmid expressing SV40 large T antigen. Full-length *Listeria* EF-Tu was used to generate α-EF-Tu antisera (*Archambaud et al., 2005*). We used α-ISG15 antibodies from Abcam, UK (ISG15 3E5, ab48020), from Cell Signaling, Danvers, Massachusetts (2743S) and from Santa Cruz, Dallas, Texas (ISG15 F-9, sc-166755). We used an α-Reticulon 4 antibody from Santa Cruz (RTN4 Nogo4 C-4, sc-271878) and an α-Atlastin-3 antibody from Millipore, Billerica, Massachusetts (ABT20). We used α-TRIM25 (ab167154) and α-UBE2L6 (ab115524) antibodies from Abcam. We used an α-ACTIN antibody from Sigma, Saint Louis, Missouri (AC-15, A5441). For immunofluorescence, we used an α-TGN46 antibody from Abcam (ab50595), an α-PDI antibody from Stressgen, Enzo Lifesciences, Farmingdale, New York (PDI (1D3) ADI-SPA-891-F) and an α-GM130 antibody from BD Transduction Laboratories, Franklin Lakes, New Jersey (610823). We used recombinant B18R (34-8185-81) and ELISA kits for human IL-8 (88-8086-88), human IL-6 (88-7066-88) and mouse IL-6 (88-7064-88) from eBiosciences, San Diego, California. We used a human Pro-ISG15 ELISA kit from EMD Millipore (CBA107) and a human cytokine array from Signosis, Sunnyvale, California (EA-4002).

### Bacterial and mammalian growth conditions and infections

Bacterial strains used in this study are shown in *Table 1*. Mutant strains of *Listeria* generated for this study were deleted from start to stop codon leaving intergenic regions intact (BUG3646, BUG3647, and BUG3648). The triple mutant (LLO, PLCA, PLCB) was constructed sequentially starting from a single mutant (as opposed to a deletion of the entire operon). Genes were deleted in *Listeria* strain EGD-e prfA* (BUG 3057) by double recombination as previously described (*Arnaud et al., 2004*). *Listeria* strains and *S. aureus* were grown in brain–heart infusion media (BD). *S. typhimurium* was grown in Luria broth and *S. flexneri* in Tryptic Soy broth (BD). Prior to infection overnight cultures of bacterial strains were diluted in new media and grown to exponential phase (OD 0.8 to 1), washed three times in serum-free mammalian cell culture media and resuspended in mammalian cell culture media at the indicated MOI. A fixed volume was then added to each well. Cells were centrifuged for 1 min at 1000 rpm to synchronize infection. The cells were then incubated with the bacteria for 1 hr at 37°C, 10% $CO_2$. Following this incubation, the cells were washed with room temperature 1× PBS and cell growth medium with 10% serum was added with 20 μg/ml gentamicin to kill extracellular bacteria. The cells were then harvested at the time point indicated in each figure.

   HeLa cells (ATCC, CCL-2) were grown in MEM with Glutamax (Gibco, Waltham, Massachusetts) supplemented with 10% fetal bovine serum, non-essential amino acids (Gibco) and sodium pyruvate

**Table 1.** Bacterial strains used in this study

| Bacterial species | Strain |
| --- | --- |
| *Listeria monocytogenes* | EGD BUG600 |
| *Listeria monocytogenes Δhly* | EGD BUG2132 |
| *Listeria monocytogenes ΔactA* | EGD BUG2140 |
| *Listeria innocua* | Clip11262 BUG499 |
| *Listeria innocua* (InlB) | Clip11262 BUG1642 |
| *Staphylococcus aureus* | SH1000 |
| *Salmonella typhimurium* | SR-11 |
| *Shigella flexneri* | M90T |
| *Listeria monocytogenes* | EGD-e PrfA* BUG3057 |
| *Listeria monocytogenes ΔplcA* | EGD-e PrfA* BUG3646 |
| *Listeria monocytogenes Δhly* | EGD-e PrfA* BUG3647 |
| *Listeria monocytogenes Δhly ΔplcA ΔplcB* | EGD-e PrfA* BUG3648 |
| *Listeria monocytogenes* | EGD-e BUG1600 |

(Gibco). 2fTGH cell lines and MEFs were grown in DMEM with 2 mM Glutamax supplemented with 10% fetal bovine serum. When indicated primary MEFs were transformed with a plasmid expressing SV40 Large T antigen (BUG 3790). Prior to infection, cells were seeded to attain 80% confluence on the day of infection.

## In vivo infections

Female C57BL/6 mice (Isg15+/+ or Isg15−/−) were infected intravenously between 8 and 12 weeks of age with $5 \times 10^5$ bacteria per animal and sacrificed 72 hr following infection. Colony forming units per organ (liver or spleen) were enumerated after tissue dissociation and serial dilutions in sterile saline. For the SDS-PAGE of animal samples, wild-type mice were infected intravenously with *Listeria* or injected with sterile saline solution and sacrificed after 72 hr. The liver and spleen were isolated and dissociated. Tissue homogenates were then centrifuged at 14,000 rpm for 10 min at 4°C. For liver tissue, an aliquot of the soluble fraction below the layer of fat was removed and resuspended in 2× Laemmli buffer. For the spleen an aliquot of the supernatant was resuspended in 2× Laemmli buffer. The samples were then run on an SDS-PAGE gel and blotted for ISG15 levels.

## Ethics statement

This study was carried out in strict accordance with the French national and European laws and conformed to the Council Directive on the approximation of laws, regulations and administrative provisions of the Member States regarding the protection of animals used for experimental and other scientific purposes (86/609/Eec). Experiments that relied on laboratory animals were performed in strict accordance with the Institut Pasteur's regulations for animal care and use protocol, which was approved by the Animal experiment Committee of the Institut Pasteur (approval number n°03–49).

## SILAC and His-affinity purification

For SILAC labeling, DMEM without L-Lysine, L-Arginine, and L-Glutamine (Silantes, Germany) was supplemented with 10% dialyzed serum (Invitrogen, Waltham, Massachusetts), 2 mM Glutamax (Invitrogen) and naturally occurring L-Lysine HCl and L-Arginine HCl ($K_0R_0$, Light condition, Sigma), 4,4,5,5-$D_4$-L-Lysine HCl and $^{13}C_6$-L-Arginine HCl ($K_4$,$R_6$, Medium condition, Silantes) or $^{13}C_6$,$^{15}N_2$-L-Lysine HCl and $^{13}C_6$,$^{15}N_4$-L-Arginine HCl ($K_8$, $R_{10}$, Heavy condition, Silantes). Lysine was added at its normal concentration in DMEM (146 mg/l); however, arginine was added at 30% of its normal concentration (25 mg/l) to prevent metabolic arginine to proline conversion. Stably transduced cell lines were labeled for a minimum of seven passages in their respective medium prior to treatment. Control cells (pBabe plasmid empty vector) were labeled light while Flag-His ISG15 ectopic expression cells (pBabe-Flag-His ISG15 mature) were labeled medium or heavy.

The nickel affinity purification was performed as described previously (*Tatham et al., 2009*), but adapted for mass-spectrometry (*Impens et al., 2014*). We expanded the SILAC labeled cells to two large 500 cm$^2$ per condition (approximately $10 \times 10^7$ cells per condition). We then treated the heavy-labeled ISG15-expressing cells with type I interferon (interferon α2) at 1000 units per ml for 40 hr while the medium and light-labeled cells were not treated. At 40 hr, post-treatment cells were lysed in 8 ml lysis buffer per 500-cm$^2$ dish (6 M Guanidium-HCl, 10 mM Tris, 100 mM sodium phosphate buffer pH 8.0). The lysates from two dishes were combined for each condition, sonicated and centrifuged. The pellet was discarded and the protein concentration in the lysate was measured using a Bradford assay (Biorad, Hercules, California). Equal protein amounts of each condition were mixed and proteins were reduced and alkylated by incubation with 5 mM tris(2-carboxyethyl)phosphine (TCEP) and 10 mM chloroacetamide for 30 min at 37˚C in the dark. Excess chloroacetamide was quenched with 20 mM dithriothreitol prior to incubation of the lysates overnight on a rotating wheel at 4˚C with 1 ml of packed NiNTA agarose beads that were pre-equilibrated in lysis buffer (Qiagen, Netherlands). The following day, the agarose beads were washed once in lysis buffer supplemented with 0.1% Triton X-100 and 5 mM β-mercaptoethanol. They were then washed once in pH 8.0 wash buffer (8 M Urea, 10 mM Tris, 100 mM sodium phosphate buffer pH 8.0, 0.1% Triton X-100, 5 mM β-mercaptoethanol), three times in pH 6.3 wash buffer (8 M Urea, 10 mM Tris, 100 mM sodium phosphate buffer pH 6.3, 0.1% Triton X-100, 5 mM β-mercaptoethanol), and eluted in 1.5 ml 100 mM sodium phosphate buffer pH 6.8, 200 mM imidazole for 20 min at room temperature. The eluate contained approximately 500 µg of protein and was further analyzed by LC-MS/MS.

## LC-MS/MS

The eluate from the His-affinity purification was further diluted with 8.5 ml 50 mM ammonium bicarbonate and proteins were digested with 20 µg trypsin overnight at 37˚C (Promega, Madison, Wisconsin). Peptides were then purified on a Sep-Pak C18 cartridge (Waters, Milford, Massachusetts), and 2 µg was injected for LC-MS/MS analysis on an Easy-nLC 1000 UHPLC system (Thermo Fisher Scientific, Waltham, Massachusetts) in line connected to a Q Exactive mass spectrometer with a NanoFlex source (Thermo Fisher Scientific). The sample was loaded on a reverse-phase column (made in-house, 75 µm I.D. × 300 mm, 1.9 µm beads C18 Reprosil-Pur, Dr Maisch GmbH, Germany) placed in a column oven (Sonation GmbH, Biberach, Germany) maintaining a constant temperature of 55˚C. Peptides were eluted by a linear increase from 5 to 28% acetonitrile in 0.1% formic acid over 130 min followed by a 55 min linear increase to 45% acetonitrile in 0.1% formic acid at a constant flow rate of 250 nl/min. The mass spectrometer was operated in data-dependent mode, automatically switching between MS and MS/MS acquisition for the 15 most abundant ion peaks per MS spectrum. Full-scan MS spectra (300–1800 m/z) were acquired at a resolution of 70,000 after accumulation to a target value of 1,000,000 with a maximum fill time of 120 ms. The 15 most intense ions above a threshold value of 100,000 were isolated (window of 2.5 Th) for fragmentation by CID at a normalized collision energy of 25% after filling the trap at a target value of 500,000 for maximum 120 ms with an underfill ratio of 2.5%. The S-lens RF level was set at 55, and we excluded precursor ions with single, unassigned, and charge states above eight from fragmentation selection.

Data analysis was performed with MaxQuant (version 1.4.1.2) (*Cox and Mann, 2008*) using the Andromeda search engine (*Cox et al., 2011*) with default search settings including a false discovery rate set at 1% on both the peptide and protein level. Spectra were searched against the human proteins in the Uniprot/Swiss-Prot database (database release version of January 2014 containing 20,272 human protein sequences, www.uniprot.org) with a mass tolerance for precursor and fragment ions of 4.5 and 20 ppm, respectively, during the main search. To enable the identification of SILAC-labeled peptides, the multiplicity was set to three with Lys$_4$ and Arg$_6$ settings in the medium channel and Lys$_8$ and Arg$_{10}$ in the heavy channel, allowing for a maximum of 3 labeled amino acids per peptide. Enzyme specificity was set as C-terminal to arginine and lysine, also allowing cleavage at proline bonds and a maximum of three missed cleavages. Variable modifications were set to GlyGly modification of lysine residues, oxidation of methionine residues, and pyroglutamate formation of N-terminal glutamine residues. Carbamidomethyl formation of cysteine residues was set as a fixed modification. In total, 692 proteins were quantified, and for each protein, the log$_2$ values of the normalized heavy/medium and normalized medium/light ratios were plotted against each other to generate the scatter plot depicted in *Figure 4*, proteins listed in *Figure 4E* and *Figure 4—source data 1*. Proteins with log$_2$ medium/light ratios >1 were considered to be upregulated upon ISG15

expression, while proteins with $\log_2$ heavy/medium ratios >1 were classified to be upregulated after interferon treatment. The mass spectrometry proteomics data have been deposited to the ProteomeXchange Consortium (*Vizcaino et al., 2014*) via the PRIDE partner repository with the dataset identifier PXD001805. GO terms enrichment analyses were performed using Database for Annotation, Visualization and Integrated Discovery (DAVID) bioinformatics resources (*Huang da et al., 2009*).

## qRT-PCR, RNASeq, and SDS-PAGE

Total RNA was isolated using an RNAeasy kit (Qiagen). 1 µg of RNA from each condition was reverse transcribed using the iScript cDNA synthesis kit (Biorad), and qRT-PCR was performed using SsoFast EvaGreen supermix (Biorad) on a CFX384 Real-Time System with a C1000 Touch Cycler (Biorad). Data were then analyzed using the ΔΔCT method relative to GAPDH levels for each sample.

For RNASeq uninfected LoVo cells were compared to LoVo cells infected with *L. monocytogenes* strain EGD-e for 24 hr. The experiment was performed in triplicate, and total RNA following infection was isolated using an RNeasy kit (Qiagen). Samples were subsequently checked for quality and sent to Fasteris for library preparation and analysis. The RNASeq data has been uploaded to Array Express with the accession E-MTAB-3649 (*Radoshevich et al., 2015b*; https://www.ebi.ac.uk/arrayexpress/experiments/E-MTAB-3649/) entitled 'ISG15 counteracts Listeria monocytogenes infection'.

For SDS-PAGE analysis, samples were lysed following one wash in 1× PBS with 2× Laemmli buffer. The cells were scraped in the sample buffer, resuspended and boiled at 95°C for 5 min. Prior to running the samples on a gel, they were sonicated two times for 8 s each to disrupt DNA. For SDS-PAGE following a nickel-affinity purification, proteins were eluted with the following buffer (200 mM imidazole, 5% SDS, 150 mM Tris-HCl pH 6.7, 30% glycerol, 720 mM β-mercaptoethanol, and 0.0025% bromophenol blue). Gels were transferred using an iBLOT transfer system (Invitrogen), blocked in 5% milk for 1 hr at RT, incubated with primary antibody overnight at 4°C, washed with 0.1% Tween in 1× PBS three times (each wash for 7 min), and incubated for an hour at room temperature with secondary antibody coupled to HRP. Blots were washed again three times with 0.1% Tween in 1× PBS and revealed using ECL-2 (Pierce, Waltham, Massachusetts).

## Cloning, transfection, viral transduction, and permeabilization

Sandra Pellegrini provided us with a plasmid containing a 3XFlag-6His-tagged mature human ISG15 in pCDNA3, which was a kind gift of Jon Huibregtse. This plasmid was used to clone a 3XFlag-6His-ISG15 into pBabe-Puro (kindly provided by Jayanta Debnath) using BamHI and SalI (BUG3354). Retroviral particles were then generated by co-transfection of viral GagPol MoMLV (BUG2666) and VSV-G (BUG2667, both plasmids generously provided by Thierry Heidmann) and pBabe-puro 3xFlag-6His-ISG15 in 293T cells. Cells were transfected using Fugene HD according to manufacturer's instructions (Promega). Supernatants were collected and applied to HeLa cells in the presence of polybrene (Millipore) as described (in the protocol 'Production of retroviruses using Fugene 6' from the Weinberg lab on Addgene). Cells were then selected using 2 µg/ml puromycin for 3 days. The population of cells that survived puromycin treatment was expanded and tested for ISG15 expression. For maintenance, cells were grown without puromycin and checked periodically for ISG15 expression. In order to deliver Cyclic diAMP to the cytosol, cells were permeabilized with saponin according to the protocol described in *Johnson et al. (1996)* with the indicated concentrations of Cyclic diAMP (Biolog, Germany).

## siRNA treatment

For siRNA treatment of cells, reverse transfection was used to minimize the siRNA concentration required for knockdown. siGenome Smartpools (Dharmacon, Lafayette, Colorado) were resuspended and aliquoted as per manufacturer's instructions at a 20 µM concentration. Prior to transfection, stocks were further diluted 1:2 with RNAase-free water, and 1.5 µl siRNA pool was added to 25 µl serum-free media per well for a 24-well format. The final concentration per well was 60 nM. Equal volumes of Lipofectamine RNAiMax (Invitrogen, 1.5 µl) were added to a second tube of 25 µl serum-free media, and master mixes were made where possible to minimize pipetting error. After mixing the smart pool tube with the RNAiMax tube and incubating for 15 min, 53 µl of mix was added to each well. Subsequently, 450 µl of cell suspension was added to each well at a concentration of 222,000 cells/ml

(100,000 cells per well). For larger scale knock down, the same concentrations were used but smart pools, RNAiMax, and cell concentrations were scaled up accordingly. Since ISG15 is induced by infection, we delivered siRNA during cell seeding and infected 24 hr later to make sure that the RNAi complexes were still present in the cells while ISG15 was induced. For the siRNA screen, cells were treated with siRNA pools for 72 hr and then infected with *Listeria* using the siControl condition as a positive control for ISG15 induction and uninfected/untransfected cells as the baseline for normal levels of ISG15 (since we show that nucleic acids alone can lead to ISG15 induction). Blots were quantified using ImageJ relative to actin as a loading control.

## Immunofluorescence, microscopy, and FACS

Cells were plated on coverslips the day prior to an experiment. Cells were fixed in 4% PFA (Electron Microscopy Sciences, Hatfield, Pennsylvania) for 10 min at room temperature and permeabilized in 0.5% Triton X-100 (Sigma). Coverslips processed for immunofluorescence were mounted on microscopy glass slides using Fluoromount G (Interchim, France), and images were acquired using an inverted wide-field fluorescence microscope (AxioVert 200M, Carl Zeiss Microscopy, Germany) equipped with an EMCCD Neo camera (Andor, Ireland) and the software MetaMorph (Molecular Devices, Sunnyvale, California). FACS samples were labeled with 100 nM ER-Tracker Blue-White DPX (Life Technologies) for 30 min, trypsinized, washed, and analyzed on a BD LSRFortessa.

## Acknowledgements

We thank Drs Sandra Pellegrini, Jon Huibregtse, Gia Voeltz, Sabrina Jabs, Matthew Albert, Scott Werneke, Jayanta Debnath, and Thierry Heidmann for generously providing reagents, and Dr Sandra Pellegrini for critically reading the manuscript. We thank Edith Gouin for production of antibodies, Dr Christophe Bécavin for generating the heat map for the proteomics data and for help uploading the RNASeq data. We thank Mikael Koutero and Andrzej Prokop for help with the RNASeq data. We thank Dr Georges Azar for help with FACS. We thank Drs Nathalie Rohlion, Alessandro Pagliuso, and Fabrizia Stavru for helpful discussions and the Pasteur Proteomics Platform.

## Additional information

### Competing interests

PC: Reviewing editor, *eLife*. The other authors declare that no competing interests exist.

### Funding

| Funder | Grant reference | Author |
|---|---|---|
| Howard Hughes Medical Institute (HHMI) | Senior International Research Fellow | Pascale Cossart |
| Institut Pasteur | Bourse Roux | Lilliana Radoshevich, Francis Impens |
| Institut national de la santé et de la recherche médicale | U604 | Pascale Cossart |
| Institut national de la recherche agronomique | USC2020 | Pascale Cossart |
| Agence Nationale de la Recherche | LISTRESS; PROANTILIS | Pascale Cossart |
| Agence Nationale de la Recherche | Laboratoire d'Excellence (ANR-10-LABX-62-IBEID) | Pascale Cossart |
| European Research Council (ERC) | 233348 MODELIST | Pascale Cossart |
| Human Frontier Science Program (HFSP) | Long Term Fellow | Lilliana Radoshevich |
| European Molecular Biology Organization (EMBO) | Long Term Fellow | Lilliana Radoshevich |

| Funder | Grant reference | Author |
| --- | --- | --- |
| Fondation Le Roch les Mousquetaires | | Pascale Cossart |
| Louis-Jeannet Foundation | | Pascale Cossart |

The funders had no role in study design, data collection and interpretation, or the decision to submit the work for publication.

### Author contributions
LR, Performed the experiments, interpreted the data and wrote the manuscript with PC, Conception and design, Acquisition of data, Analysis and interpretation of data, Drafting or revising the article; FI, Performed the mass spectrometry experiments and analysis and helped to interpret data, Conception and design, Acquisition of data, Analysis and interpretation of data; DR, Established the affinity purification protocol for ubl proteins used in this project and helped to interpret data, Analysis and interpretation of data, Contributed unpublished essential data or reagents; JJQ, JP-C, Generated the triple (*hly*, *plcA* and *plcB*) mutant and analyzed the behavior of the mutants in cells, Acquisition of data, Contributed unpublished essential data or reagents; TNT, Generated the triple (*hly*, *plcA* and *plcB*) mutant, Contributed unpublished essential data or reagents; M-AN, Performed all the in vivo experiments (infection and dissection), Acquisition of data; HB, Performed the transcriptomic analysis of infected cells compared to uninfected cells, Acquisition of data, Contributed unpublished essential data or reagents; OD, Performed the transcriptomic analysis of infected cells compared to uninfected cells and helped with analysis for in vivo experiments, Acquisition of data, Analysis and interpretation of data, Contributed unpublished essential data or reagents; K-PK, Provided the *Isg15*−/− mice, contributed to the interpretation of infection results, suggested additional experiments and contributed to the final steps of manuscript preparation, Conception and design, Analysis and interpretation of data, Drafting or revising the article, Contributed unpublished essential data or reagents; PC, First roposed the project to LR, mentored LR and wrote the manuscript with LR, Conception and design, Analysis and interpretation of data, Drafting or revising the article

### Ethics
Animal experimentation: This study was carried out in strict accordance with the French national and European laws and conformed to the Council Directive on the approximation of laws, regulations and administrative provisions of the Member States regarding the protection of animals used for experimental and other scientific purposes (86/609/Eec). Experiments that relied on laboratory animals were performed in strict accordance with the Institut Pasteur's regulations for animal care and use protocol, which was approved by the Animal experiment Committee of the Institut Pasteur (approval number n°03–49).

# Additional files

## Major datasets
The following datasets were generated:

| Author(s) | Year | Dataset title | Dataset ID and/or URL | Database, license, and accessibility information |
| --- | --- | --- | --- | --- |
| Radoshevich L, Impens F, Ribet D, Quereda J, Marie-Anne Nahori, Helene Bierne, Olivier Dussurget, Pizarro-Cerda J, Knobeloch K-P, Cossart P | 2015 | ISG15 counteracts Listeria monocytogenes infection | http://proteomecentral.proteomexchange.org/cgi/GetDataset?ID=PXD001805 | Publicly avaialble in ProteomeXchange Consortium (Accession no: PXD001805). |
| Radoshevich L, Impens F, Ribet D, Quereda J, Marie-Anne Nahori, Helene Bierne, Olivier Dussurget, Pizarro-Cerda J, Knobeloch K-P, Cossart P | 2015 | ISG15 counteracts Listeria monocytogenes infection | https://www.ebi.ac.uk/arrayexpress/experiments/E-MTAB-3649/ | Publicly available at ArrayExpress (Accession no: E-MTAB-3649). |

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
