## [Decision Letter]

Thank you for sending your work entitled “ISG15 counteracts *Listeria monocytogenes* infection” for consideration at *eLife*. Your article has been favorably evaluated by Fiona Watt (Senior editor) and two reviewers, one of whom is a member of our Board of Reviewing Editors.

The Reviewing editor and the other reviewers discussed their comments before we reached this decision, and the Reviewing editor has assembled the following comments to help you prepare a revised submission.

The authors investigate the induction of ISG15 in non-phagocytic cells upon infection with *Listeria monocytogenes*. They observe IFN- and lysteriolysin-independent ISG induction via the STING pathway and a restrictive effect of ISG15 on *Listeria* in vitro and in vivo. Based on overexpression of ISG15 the authors suggest that transmembrane proteins of the secretory pathway are ISGylated and cytokine secretion is enhanced. The manuscript is interesting but several specific points need to be addressed before it can be considered for publication:

1) Effect of endogenous ISG15 versus overexpressed: Regarding the ISGylation of proteins in the secretory pathway, these data rely on the overexpression of ISG15 in the absence of IFN-stimulation, which would be required to induce the ISGylation machinery. The authors should infect cells with *Listeria* WT and *L. innocua* InlB without the overexpression of ISG15 to confirm a significant increase in IL-8 or IL-6 secretion, as detected with overexpressed ISG15. Furthermore the authors should monitor cytokine secretion upon *Listeria* infection in the ISG15-/- primary fibroblasts.

2) The most interesting findings relate to a restrictive effect of ISG15 on *Listeria*. However, the authors do not provide insight into its mode of action. It is not clear at which stage of the bacterial life cycle ISG15 acts (uptake, cytosol entry, proliferation, cellular defence), whether conjugation of ISG15 is required, nor what its mechanism of action is. Are other intracellular bacteria affected by ISG15? Are bacteria ISGylated?

3) ISG induction: subsection “Cytosolic *Listeria* induces ISG15” ‘our results reveal that ISG15 induction stems from cytosolic bacteria’. The authors provide no real evidence for this claim, other than lack of ISG production in cells infected with InlB expressing *L. innocua*. Any difference between *L. innocua* and *L. monocytogenes* could therefore cause their differential ability to induce ISG15. In fact, ISG15 induction by *L. monocytogenes* lacking hly, if anything, speaks against a cytosol-derived signal. It is likely cytosolic bacteria that do the trick, experimental evidence is lacking. Subsequent experiments of DNA transfection and STING dependent signalling does not prove the above point, since the authors do not address their original question (subsection “ISG15 is induced via the Cytosolic Surveillance Pathway (CSP) following sensing of cytosolic DNA”) to find the PAMP that is necessary in inducing ISG15.

---

## [Author Response]

*1) Effect of endogenous ISG15 versus overexpressed: Regarding the ISGylation of proteins in the secretory pathway, these data rely on the overexpression of ISG15 in the absence of IFN-stimulation, which would be required to induce the ISGylation machinery. The authors should infect cells with* Listeria *WT and* L. innocua *InlB without the overexpression of ISG15 to confirm a significant increase in IL-8 or IL-6 secretion, as detected with overexpressed ISG15*.

We performed the experiments and found that wild-type *Listeria* massively induced IL-8 and IL-6 secretion relative to non-infected cells and cells incubated with *Listeria innocua* expressing InlB. In both cases, increased IL-8 and IL-6 cytokine secretion correlated with induction of endogenous ISG15. These data have now been indicated in the text (Figure 5—figure supplement 1)

*Furthermore the authors should monitor cytokine secretion upon* Listeria *infection in the ISG15-/- primary fibroblasts.*

We did perform these experiments. However, we first transformed primary *isg15+/+ and isg15-/-* mefs with SV40 large T antigen to avoid variability from passage to passage in primary mefs. These cells secreted significantly less cytokines (IL-6) than their wild type counterparts, despite a much higher bacteria load than in wild type cells. These data are now included in Figure 5—figure supplement 1.

*2) The most interesting findings relate to a restrictive effect of ISG15 on* Listeria*. However, the authors do not provide insight into its mode of action. It is not clear at which stage of the bacterial life cycle ISG15 acts (uptake, cytosol entry, proliferation, cellular defence), whether conjugation of ISG15 is required, nor what its mechanism of action is. Are other intracellular bacteria affected by ISG15? Are bacteria ISGylated?*

Effect of ISG15 on bacterial life cycle:

To assess whether there is an effect of ISG15 on bacterial uptake, inside-out staining was performed. Briefly cells (control or ISG15-overexpressing) were infected with GFP bacteria, fixed and external bacteria were immunolabeled with a second fluorophore. Cells were then quantified for percentage of singly labeled bacteria (bacteria inside the cells). There was no difference in bacterial uptake between the two cell lines (control or ISG15 overexpressing, Figure 3—figure supplement 1). We then performed a time course of infection in cells that overexpress ISG15 to identify when ISG15 starts to counteract infection. ISG15 already has an effect three hours post infection and the ratio of bacteria relative to control cells remained the same until twenty-four hours post infection (Figure 3—figure supplement 1). This indicated an early effect of ISG15 on bacteria and excludes a restrictive effect on proliferation or an effect mediated by cellular defense. Taken together, ISG15 overexpression seems to have an early effect on infection which may correspond to vacuolar escape.

Is conjugation of ISG15 required for the effect?

Ube1L is the E1 enzyme for ISGylation so we made use of *ube1L* -/- KO mefs to determine whether conjugation is required for ISG15-mediated protection. Ube1L-deficient and ISG15-deficient primary mefs both had very high bacterial burdens compared to wildtype MEFs revealing that conjugation is critical for the protective ISG15 effect (Figure 3).

In order to confirm this data, we stably transduced HeLa cells with retrovirus expressing a conjugation-incompetent ISG15 (ISG15ΔGG) and tested whether the resulting cells were protected from *Listeria* infection as are cells that overexpress wildtype ISG15. ISG15ΔGG cells were less protected than ISG15 wildtype cells. Here again this suggests that conjugation is critical for the effect of ISG15 on *Listeria*. However, it has been technically challenging to isolate a cell line expressing this mutant. We hypothesize that ISG15 ΔGG is deleterious to cells and thus expression of ISG15 ΔGG is never as high as wild type ISG15. We thus did not include the data in the revised version of the paper. In the future, we plan to use CRISPR/Cas9 technology in hopes of avoiding expression differences between the mutant and complemented strains but were unable to design and select these lines during the two-month revision period.

Are other intracellular bacteria affected by ISG15?

We first assessed whether *isg15* -/- mefs have an increased bacterial burden for any intracellular pathogen or whether this phenotype was specific to *Listeria*. We infected SV40 transformed *isg15* -/- mefs with *Shigella* or *Salmonella* (Figure 3—figure supplement 2). Surprisingly, *Shigella* and *Salmonella* have equal bacterial counts in *isg15* +/+ and *isg15* -/- MEFs. We intend to capitalize on this fascinating observation to help us understand how ISG15 targets *Listeria* and not *Shigella* and *Salmonella* in the future.

We next tested whether other pathogens induced ISG15. Following infection with *Listeria*, *Staphylococcus aureus*, *Salmonella* and *Shigella,* only *Staphylococcus* led to as much ISG15 induction as *Listeria* infection (Figure 3—figure supplement 2). Despite similar bacterial counts, much less ISG15 is induced by the intracellular pathogens, *Shigella* and *Salmonella* than *Listeria*.

Are bacteria ISGylated?

We have analyzed which proteins are ISGylated after infection by mass spectrometry (enrichment of ISG15 following infection) or by immunofluorescence and so far have been unable to detect either ISGylated bacteria or that bacterial proteins were ISGylated.

*3) ISG induction: subsection “Cytosolic* Listeria *induces ISG15” ‘our results reveal that ISG15 induction stems from cytosolic bacteria’. The authors provide no real evidence for this claim, other than lack of ISG production in cells infected with InlB expressing* L. innocua*. Any difference between* L. innocua *and* L. monocytogenes *could therefore cause their differential ability to induce ISG15. In fact, ISG15 induction by* L. monocytogenes *lacking hly, if anything, speaks against a cytosol-derived signal. It is likely cytosolic bacteria that do the trick, experimental evidence is lacking. Subsequent experiments of DNA transfection and STING dependent signalling does not prove the above point, since the authors do not address their original question (subsection “ISG15 is induced via the Cytosolic Surveillance Pathway (CSP) following sensing of cytosolic DNA”) to find the PAMP that is necessary in inducing ISG15*.

There are reports (43; 52) and our own unpublished results that indicate that in certain human epithelial cell lines, including HeLa (which we used in this study), *Listeria monocytogenes Δhly* is able to escape into the cytosol owing to the possible action of the phospolipases plcA and plcB. We have clarified this point in the text. We thus constructed a triple mutant of *Δhly ΔplcA ΔplcB* to more directly address the reviewers concerns. This mutant was unable to escape into the cytosol whereas *ΔplcA* or *Δhly* both could (Figure 2—figure supplement 1). Interestingly, only the triple mutant did not lead to ISG15 induction, whereas the single mutants with access to the cytosol did (Figure 2). We believe that this data is a strong indication that the sensing of *Listeria* that results in ISG15 induction occurs in the cytosol.